# Evaluation of the Influence of Road Geometry on Overtaking Cyclists on Two-Lane Rural Roads

**DOI:** 10.3390/ijerph19159302

**Published:** 2022-07-29

**Authors:** Griselda López, Sara Moll, Ana María Pérez-Zuriaga, Alfredo García

**Affiliations:** Highway Engineering Research Group, Universitat Politècnica de València, Cami de Vera sn, 46022 Valencia, Spain; samolmon@upvnet.upv.es (S.M.); anpezu@tra.upv.es (A.M.P.-Z.); agarciag@tra.upv.es (A.G.)

**Keywords:** overtaking manoeuvre, cyclist group, two-lane rural road, instrumented bicycle, road geometry

## Abstract

Road cycling, both individually and in groups, is common in Spain, where most two-lane rural roads have no cycle lanes. Due to this, and the difference in speed between drivers and cyclists, the overtaking manoeuvre is one of the most dangerous interactions. This study analyses how road geometry influences the overtaking manoeuvre performance. Field data of 1355 overtaking manoeuvres were collected using instrumented bicycles, riding along different rural road segments, and considering individual, medium and large groups of cyclists. The safety variables that characterise the overtaking manoeuvre are overtaking vehicle speed and lateral clearance. These variables have been correlated to geometric characteristics of the road, such as the type of centre line, the horizontal alignment, the speed limit, and the road cross section. Regression models have been fitted considering each cyclist group size and configuration. For individuals and medium groups, wider roads generate higher lateral clearances and lower overtaking speeds, while for large groups only the solid centre line was significant, generating lower clearances and higher speeds. Results suggest that other factors need to be considered, especially for large groups. Results offer a deeper understanding of the phenomenon by providing key points for improving road geometry design, such as widening the shoulders.

## 1. Introduction

In recent decades there has been an increase in cycle traffic on rural roads, especially in Spain due to its good orographic and climatological conditions. Most of the cyclists who ride on rural roads are sport or recreational cyclists, and they tend to ride individually or in groups. The number of federal licenses for cyclists and cycling clubs in Spain was 49% higher in 2020 than in 2008 [1]. These sport cyclists have to share the road with motorised vehicles, as most two-lane rural roads are designed only for motorised vehicles without specific infrastructure for cyclists. Therefore, these road users have to interact usually in overtaking manoeuvres due to the speed differences between them. The overtaking manoeuvres is a safety issue, especially on two-lane rural roads, where accident data presents a higher severity than on the urban environment [2].

The overtaking manoeuvre between drivers and cyclists has been analysed in several studies. Most of these studies focus on safety measures during the overtaking manoeuvre, with the lateral clearance between the vehicle and the bicycle as the main safety measure analysed. Rubie et al. [3] conducted a complete literature review on lateral clearance when motor vehicles overtake bicycles. They analysed more than 40 articles and concluded that there were significant positive relationships between lateral distance and road width and between lateral distance and speed limit, and that there was a smaller lateral distance when cyclists were overtaken by buses rather than cars. They also observed inconsistent results on on-road bicycle infrastructure, gender and type of cyclists, and overtaking strategy (flying or accelerative) on lateral distance with respect to the research analysed.

Regarding the influence of roadway geometry on lateral clearance, there is some research. Chapman and Noyce [4] performed an analysis of 1151 observations of overtaking manoeuvres using an instrumented bicycle riding on rural roads in Wisconsin. The authors completed a regression analysis considering the influence of several road geometric factors on lateral clearance. The results of their model showed that geometric elements, such as roadway grade, shoulder presence and width, marked centre line, and roadway design speed (posted speed limit) significantly affect how drivers use a rural roadway, especially when overtaking a bicyclist. Other nongeometric factors such as bicycle speed and oncoming vehicle presence also affected lateral clearance. García et al. [5] analysed the effect of road geometry on the interaction between cyclists and motor vehicles on Spanish two-lane rural roads. They used an instrumented bicycle riding on seven rural roads with different geometric conditions. They not only analysed the lateral clearance, but also the speed of the overtaking vehicle. They observed higher lateral clearances on wider rural roads, considering the lane and shoulder width. The lateral clearance was higher on left curves and lower on right curves than in tangents. They also obtained a non-clear relation between the speed of the overtaking vehicle and the lane and shoulder width. Recently, Bella and Silvestri [6] used a driving simulator to analyse the effect of some geometric features on the lateral clearance between drivers and cyclists. They analysed the effect of the road cross-section and the horizontal alignment on the overtaking manoeuvre to one cyclist riding individually. Their results stated that wider bicycle lanes ensured a higher lateral clearance between driver and cyclists, and lower lateral clearances were recorded at tangent elements than at curves. On the left curve, the highest lateral clearance was recorded.

All these studies have focused on overtaking manoeuvres with cyclists riding individually. However, in Spain it is very common to find cyclists riding in groups (with different sizes), and in different configurations (either in-line or two-abreast). There are very few studies that consider groups of cyclists riding on two-lane rural roads, and most of them are focused on group dynamics in cycling racing environments [7]. Fraser and Meuleners [8] analysed the risk factors associated with unsafe events involving a motor vehicle and a group of cyclists. They stated that roads with a speed limit higher than 60 km/h increase the risk of an unsafe event significantly. They also concluded that cyclist groups riding two-abreast in the traffic lane significantly reduced the risk of an unsafe event compared to riding in a single file in the traffic lane. Lopez et al. [9] analysed the objective and subjective perception of risk in the overtaking of cyclists riding in groups on rural two-lane roads in Spain. Recently, another study performed in Spain by Pérez-Zuriaga [10] presented a descriptive analysis to explore driver behaviour when overtaking cyclists riding in different group configurations. Overtaking manoeuvres were evaluated by analysing the lateral distance, the speed, and other characteristics of the manoeuvre.

Nevertheless, this previous research considering cyclist groups is limited, and the effect of road geometric features on overtaking groups of cyclists has not been investigated. This paper presents an in-depth analysis of the influence of geometry on overtaking cyclists taking into account these characteristics (group size and configuration); with the safety measures modelling the lateral clearance between the overtaking vehicle and the bicycles, and the overtaking vehicle speed during the manoeuvre.

It is clear that there are other factors that have a clear influence on driver behaviour when overtake cyclists on rural roads, such as traffic, vehicle and human factors. In this study, only the influence of factors related to road geometry on the way in which drivers overtake cyclists has been explored. A novelty of this study is the analysis of overtaking manoeuvres to cyclists riding individually and in medium and large groups. The results obtained will allow us to gain an in-depth understanding of the phenomenon, and to recommend improvements to road geometry elements in order to achieve safer infrastructures considering all road users.

## 2. Materials and Methods

The interactions between drivers and groups of cyclists were analysed on five two-lane rural roads using instrumented bicycles. Then, the influence of the road geometry on the safety variables related to the overtaking manoeuvre was analysed. Models for lateral clearance and overtaking vehicle speed were fitted considering factors related to the road geometry.

### 2.1. Data Collection

Data collection was performed on five two-lane rural roads located in Valencia (Spain), with a longitudinal slope lower than 4% and a percentage of heavy goods vehicles lower than 2%, being its influence on vehicles operation minimal. Geometric and traffic characteristics of these segments are shown in Figure 1.

The bicycles were equipped with Garmin Virb Elite small high-definition video cameras with an integrated GPS. The first and last bicycle of the group were equipped also with a laser device to register the lateral distance and the relative speed between the vehicle and the bicycle during the overtaking manoeuvre (Figure 2 left).

To analyse the effect of cyclist group configuration in the overtaking manoeuvre, a total of five configurations of cyclist groups were considered, varying in size and the configuration, in-line (L) or two-abreast (TA) (see Figure 2 right).

The cyclists who participated in the experiment were cyclists with extensive experience in road cycling and group riding. Cyclists on Spanish roads have to ride on the shoulder or on the right edge of the road if the shoulder is impracticable [11]. The cyclists who participated in the study rode according to these standards, but in a naturalistic way. On wider roads, cyclists rode on the shoulder, but on narrow roads they rode in the lane.

Based on previous observations of cycle demand on these roads, a data-collection schedule was designed. Data collection was conducted on weekday mornings for individual cyclists and medium-sized groups of cyclists, whereas for the large group of 10 cyclists, data were collected on weekend mornings, as these are the days when drivers expect to interact with large groups. Data were collected in 24 sessions due to the difficulty of moving all the material and human equipment necessary to carry out the tests to the road segments. The cyclists participating in the test completed round trips on the 5 road segments and rode in the 7 group configurations indicated.

### 2.2. Data Reduction

For each overtaking manoeuvre, data were collected from different data sources: the videos cameras and the laser device.

The video camera located on the rear of the bicycle recorded the approach and initial phase of the overtaking manoeuvre, whereas the second camera, located on the handlebar of the bicycle, recorded the last phase of the overtaking manoeuvre, when the vehicle returns to its lane. GPS incorporated into the cameras recorded the speed and position of the cyclists at any moment, especially during each overtaking manoeuvre. The videos were reviewed with a specific software that allows the GPS data to be incorporated inside the video review. From the video review, several variables were obtained for each overtaking manoeuvre, such as the centre line type, the lane and shoulder width, and the opposing lane invasion where the overtaking manoeuvre was performed. The horizontal alignment of the studied road segments was estimated with a specific software and, thanks to the GPS information of the cameras, the horizontal element where each overtaking manoeuvre was performed was identified. The independent variables related to road geometry considered in this study are shown in Table 1.

The laser device registered the relative speed and the lateral distance between the vehicle and the bicycle in each overtaking manoeuvre. In this study, the surrogate measures of safety, overtaking vehicle speed (Sv), and lateral clearance (Lc) were used as dependent variables. Sv was calculated by the relative speed registered by the laser device plus the bicycle speed registered by the GPS of the cameras, and Lc was calculated from the lateral distance recorded by the laser device by subtracting half bicycle handlebar and the rear mirror of the motorised vehicle.

### 2.3. Model Development

Before developing the models, it is necessary to first transform the qualitative variables. Variables with two and three categories were transformed into 1 dummy variable and 2 dummy variables, respectively. Of the variables formed by 2 dummies, only one is included in the multiple regression analysis, while 2 of the 3 dummies are included in the model, leaving the remaining ones with a reference category. Then, the following transformations were performed:Centre-line with 2 categories was transformed into 1 dummy variable: solid line (included into the models), with dashed line being the reference category.Horizontal alignment with 3 categories was transformed into 2 dummy variables: tangent and right curve (included in the models), with left curve being the reference category.Opposing lane invasion with 2 categories was transformed into 1 dummy variable: invasion (included into the models), with no invasion being the reference category.Configuration with 2 categories was transformed into 1 dummy variable: in-line (included into the models), with two-abreast being the reference category.

It is important that the predictor variables included in the regression analysis are not too highly correlated with each other. To avoid multicollinearity problems, a correlation analysis before the regression analysis was conducted. Highly correlated variables (Pearson coefficient > 0.7 and < −0.7) were not included in the models.

Multiple regression models were developed following the stepwise regression method, finding the factors that significantly influence the lateral clearance and speed of overtaking vehicles and selecting the variables with the highest goodness of fit (adjusted R^2^). After determining the predictors, the ordinary least-squares method was performed to establish the regression model and estimate the parameters based on the minimum square sum of the relative error. To determine the significance of the model and each factor, the F-test and *t*-test were employed, respectively. The significance level of all test methods was set as 0.05. In addition, Durbin–Watson (DW) test was also applied to test the serial correlation with a criterion of DW ≈ 2.

## 3. Results

This section describes the results of the study. First, a descriptive analysis of the data regarding cyclist group configurations and road geometric variables was performed. Next, models of the lateral clearance and speed of overtaking vehicles for each cyclist group size were developed and fitted.

### 3.1. Data Description

Only overtaking manoeuvres performed by passenger cars were considered; then, a total of 1355 overtaking manoeuvres were registered. In all road segments, a higher number of overtaking manoeuvres were registered when one cyclist rode individually because his lower dimension. When the cyclist group was growing, the number of overtaking manoeuvres registered were lower, since it is more difficult to overtake a larger group because more overtaking time is needed (Figure 3).

### 3.2. Lateral Clearance and Overtaking Vehicle Speed

Table 2 shows the results of lateral clearance and overtaking vehicle speed obtained from field observations and each factor considered in this study. These results are presented for each cyclist group size (1 cyclist, 4 cyclists, and 10 cyclists) and considering different variables.

Regarding the lateral clearance, it was higher when the group consisted of more cyclists and they rode in-line, whereas for speed, the group of 10 cyclists presented the highest and the lowest vehicle overtaking speed when riding in-line and two abreast, respectively.

There were factors with a clear effect on both clearance and speed, such as the road centre line, which presented lower speed but lower clearance when it was a solid line for the three cyclist group sizes; or the opposing lane invasion, with higher clearance and speed for larger invasions. Moreover, there were other factors with no clear effect on lateral clearance and overtaking speed, such as the width of the shoulder or the speed limit.

### 3.3. Model Development and Results

With the aim of analysing the influence of road geometry on the overtaking manoeuvre to cyclists, multiple regression models incorporating geometric factors were developed. Lateral clearance and overtaking vehicle speed were used as dependent variables to develop the models.

Before establishing the regression models, normality of the data distribution was analysed for the two dependent variables. In this study, the normal probability plot was used to determine the normal distribution of the data sets, as it is able to show the possible skewness of the data. Figure 4 shows the resulting distributions, as can be seen that the values of lateral clearance (Lc) and overtaking vehicle speed (Sv) were almost normal, with a slight bias towards the tails. These biases are in correlation with the phenomenon analysed, since when the driver tends to overtake the cyclist by moving farther away (or closer), it results in a slightly wider distribution at the extremes, as found in previous studies [4]. The same biases are observed with the extreme values of overtaking speed.

In order to avoid multicollinearity problems in the regression models, a correlation analysis was performed. The strongest correlations (Pearson’s coefficients greater than 0.7) were presented between the following variables: Lane width (Lw) and Speed limit (SL); Lane width (Lw) and Lane + Shoulder width (LSw); and Tangent (T) and Left curve (LCur).

Next, multiple regression models were developed. Correlated variables SL, LSw, and LCur were not considered in the development of the models. Models were estimated using the STATGRAPHICS software, adopting the stepwise regression method, which only adds to the model those variables whose influence is statistically significant (with a confidence level of 95%), and which increase the coefficient of determination (i.e., improve the percentage of explanation of the variance of the dependent variable). Then, only significant variables were drawn in the results.

First, general models have been fitted for the lateral clearance and speed of the overtaking vehicle considering, in addition to the geometrical factors already explained, a variable representing the grouping of cyclists (named Individual, being 1 when cyclists ride individually and 0 when they ride in a group). The results of these global models are shown in Table 3. It is observed that whether cyclists ride individually or in groups is a significant factor in the lateral clearance model, with higher clearances when cyclists ride individually. However, in the overtaking vehicle speed model, whether cyclists ride individually or in groups is not significant.

However, these general models do not take into account the size of the group of cyclists and their in-line or two-abreast configuration. In view of these results, specific models have been fitted for each size of cyclist group overtaken, and the models for the cyclist groups have incorporated the configuration variable, which indicates whether cyclists ride in-line or two-abreast.

Table 4 shows the results of the models developed for the dependent variable lateral clearance considering the different cyclist groups analysed. According to the goodness of fit (adjusted R^2^), the model developed for individual cyclists (M1) presents the highest value (adjusted R^2^ = 0.3185); thus, the model can explain 31.85% of the variability of the lateral clearance of overtaking to individual cyclists. The model for medium-sized groups of cyclists (M2) achieves an adjusted R^2^ = 0.2164, whereas the model for the largest groups of cyclists (M3) has the lowest fit value (R^2^ = 0.1789).

M1 refers to lateral clearances when overtaking a cyclist riding individually. The model was fitted by completing three iterations. Finally, only three factors were significant in this model. Lane and shoulder width had a significant effect on the lateral clearance the driver leave from the cyclist. Higher lane and shoulder widths are related to higher lateral clearances, with a coefficient for the lane width (Coef. = 0.4454) and for the shoulder width (Coef. = 0.1144). The third factor with significant influence in the lateral clearance was the opposing lane invasion. When the driver invaded the opposing lane (Coef. = 0.5383), the lateral clearance was higher. In this model, horizontal alignment and road centre line have no significant effect on lateral clearance (*p*-values > 0.05).

M2 corresponds to the lateral clearance when overtaking a group of four cyclists. Four iterations were needed to fit the model. The factors with a significant effect on lateral clearance were lane (Coef. = 0.5454) and shoulder width (Coef. = 0.1654), opposing lane invasion (Coef. = 0.5387), and the configuration (Coef. = 0.2348) in which cyclists ride. Higher lane and shoulder widths were related with higher lateral clearances. Higher lateral clearances were obtained when the overtaking vehicle invades the opposing lane. Finally, the in-line or two-abreast configuration in which the cyclist group rode had a significant effect on lateral clearance, with higher clearances when cyclists rode in-line. In this case, the effect of horizontal alignment and the centre line of the road had no significant effect on lateral clearance (*p*-values > 0.05).

Finally, the M3 was developed for a group of 10 cyclists. The model was fitted by completing five interactions. Three factors had a significant effect on the lateral clearance. The first was the group configuration, with higher lateral clearances when cyclists rode in-line than two-abreast (Coef. = 0.1695). The road centre line also had a significant effect, with lower clearances obtained when overtaking was performed with a solid line (Coef. = −0.1125). Finally, the opposing lane invasion had a significant effect, presenting higher clearances when the opposing lane was invaded. In this case, the horizontal alignment and the road cross section had not a statistical effect on the lateral clearance (*p*-values > 0.05).

Results of the models developed for the dependent variable (overtaking vehicle speed) and the different cyclist groups are presented in Table 5. The model developed for individual cyclists has the highest fit value (adjusted R^2^ = 0.2752), followed by the model for medium-sized groups of cyclists (adjusted R^2^ = 0.2108), whereas, like the models for lateral clearance, the model for the largest groups of cyclists has the lowest fit value (R^2^ = 0.0855).

The speed of the overtaking vehicle during the overtaking manoeuvre to one cyclist riding individually was modelled in the M4. Three iterations were required to fit the model. Factors with a significant effect on overtaking speed were lane and shoulder widths, and road centre-line type. Wider roads, with higher lane (Coef. = 53.936) and shoulder widths (Coef. = 10.687) presented higher overtaking speeds. Overtaking manoeuvres performed with a solid line (Coef. = −5.681) were performed at lower speeds. Horizontal alignment and invasion of the opposing lane had no significant effect on the speed of the overtaking vehicle (*p*-values > 0.05).

M5 corresponds to the speed of the overtaking vehicle when overtaking a group formed by four cyclists. M5 was fitted by completing two iterations. Five factors had a significant effect on the speed of the overtaking vehicle. Higher lane (Coef. = 31.925) and shoulder widths (Coef. = 7.501) generated higher speeds. Overtaking manoeuvres with opposing lane invasion (Coef. = 4.869) presented higher overtaking vehicle speeds. When a solid line (Coef. = −5.425) was present, the overtaking speed was lower. Finally, when the group of cyclists rode in-line, the speed of the overtaking vehicle was higher (Coef. = 3.008).

M6 refers to the speed of the vehicle when overtaking a group formed by 10 cyclists. In this case, the model was fitted by realising five iterations. Only two factors had a significant effect on the speed of the overtaking vehicle (*p*-values < 0.05). When the vehicle overtook with solid road centre line (Coef. = −5.162), the speed was lower. The other factor with a significant effect was the configuration of the group of cyclists. When cyclists rode in-line (Coef = 5.831), a higher speed of the overtaking vehicle was obtained. Factors related to horizontal alignment, opposing lane invasion, or cross-section width had no significant effect on the speed of the vehicle when overtaking a group of 10 cyclists (*p*-values > 0.05).

Finally, residuals of the models were checked. The DW test statistic can range from 0 to 4, and a value of 2 suggests that the residuals are uncorrelated. All the fitted models present values within these ranges. To determine the assumption regarding residuals, the plot of standardised residual vs. predicted values of lateral clearance and overtaking vehicle speed was drawn (see Figure 5). The results of the residual analysis are optimal because the data points in the scatter plot of the individual values versus the fitted (or predicted) values are similar to a random series of points scattered around zero. Overtaking manoeuvres are critical interactions between drivers and cyclists, and sometimes the speed of the overtaking vehicle and the lateral clearance between drivers and cyclists reach critical values that make these manoeuvres more dangerous. These extreme values generate some outliers that cannot be eliminated from the analysis because they are also representative of this phenomenon. However, the record of extreme values was minimal, with the standardised residual plots showing generally acceptable results.

## 4. Discussion

The influence of road geometric factors on the overtaking manoeuvre to a group of cyclists was analysed. Lateral clearance and overtaking vehicle speed were modelled by fitting two multiple regression models. Results were obtained considering three sizes of the group of cyclists: one cyclist, a medium-size group formed by four cyclists, and a large-size group of ten cyclists.

Lane and shoulder widths had a significant effect on lateral clearance and overtaking vehicle speed for overtaking manoeuvres to one cyclist and a medium group of four cyclists. In these cases, higher lateral clearances and higher vehicle speeds were related to wider roads considering both lane and shoulder widths. The effect of the lane width was higher than the shoulder width on both independent variables.

For the larger group formed by 10 cyclists, the road cross-section width had no significant effect on lateral clearance and overtaking vehicle speed. When a group of 10 cyclists was overtaken, the highest lateral clearance mean was registered. This result indicates that drivers leave more lateral clearance when overtaking large groups of cyclists, regardless of the road cross-section. However, despite having no significant effect on clearance and speed, wider roads had higher mean clearance values than narrow roads for groups of 10 cyclists. Other studies also recommend wider roads to ensure safer overtaking to cyclists [4,6,12,13]. Shackel and Parkin [14] conducted a study on how road markings and lane width influence proximity and speed of vehicles overtaking cyclists. Although their study was performed in an urban environment, they also obtained higher overtaking speeds and higher passing distances on wider lanes.

Road centre-line type had a significant effect on overtaking vehicle speed for all cyclist group sizes analysed, with lower speeds when overtaking manoeuvres were performed with a solid line. On the other hand, overtaking manoeuvres performed with a solid line only had a significant effect on lateral clearance for the larger group of 10 cyclists, resulting in lower lateral clearances when a solid line was present. These results can be related to the lower visibility available when a solid centre line is present.

The horizontal alignment where the overtaking manoeuvre was performed had no significant effect on any of the models developed. However, higher lateral clearances were obtained on left curves for one cyclist and medium size groups. This result agrees with Bella and Silvestri [6] and García et al. [5], who also obtained higher lateral clearances when overtaking manoeuvres were performed on left curves elements. However, most of the overtaking manoeuvres were performed on tangent elements, as can be seen in Table 2; this result coincides with García et al. [5], who also performed a naturalistic study using an instrumented bicycle but only considering one cyclist riding individually.

The opposing lane invasion is related with the interaction with oncoming traffic and consequently with the frontal collision risk with opposing vehicles. García et al. [5] stated that the opposing lane invasion depends on the road width and the opposing vehicles frequency, resulting in higher percentage in opposing lane invasion when the road width decreased and the opposing vehicle frequency was lower. In this study, the opposing lane invasion significantly affects the lateral clearance for all cyclist group sizes analysed. For overtaking manoeuvres with opposing lane invasion, the lateral clearance was higher, with the coefficient being similar for all the cyclist group sizes. Regarding the speed of the overtaking vehicle, only for the medium size group there was a significant effect of the invasion, resulting in a higher speed when the opposing lane was invaded. Although the effect of opposing lane invasion is not significant on the overtaking vehicle speed for 1 and 10 cyclist groups, Table 2 shows that for the shorter groups (1 and 4 cyclists) invading the opposing lane implied a higher speed, whereas for the larger group the speed was lower. This may be due to the fact that when overtaking longer groups, drivers make sure that there is no oncoming vehicle and therefore allow themselves a lower speed when overtaking in the opposing direction.

Finally, the in-line or two-abreast configuration in which the cyclist group rode was significant in all cyclist groups analysed and for both dependent variables. When cyclist groups rode in an in-line configuration, a higher lateral clearance and higher overtaking vehicle speed was obtained. These results are related with the fact that cyclists riding two-abreast occupy a higher volume than when riding in-line. Therefore, the lateral clearance is lower. Fraser and Meuleners [8] concluded that cyclists riding two-abreast significantly reduced the risk of an unsafe event compared to riding single-file. Therefore, although presenting lower lateral clearances, riding two-abreast is considered safer because of the lower overtaking vehicle speed and the increase in the visibility of the cyclists on the road.

Finally, the main contributions of this research are summarised: (i) For individual and medium groups of cyclists, wider roads presented higher lateral clearances and higher overtaking speeds; (ii) groups of cyclists riding in-line presented higher clearances and overtaking speeds; (iii) the effect of the solid centre line is related with lower overtaking speeds and lower lateral clearance in large groups.

## 5. Conclusions

In this study, the influence of the factors related to road geometry on the overtaking manoeuvres to cyclists was analysed. The safety measures used to evaluate the overtaking manoeuvres were lateral clearance and overtaking vehicle speed. Different sizes of groups of cyclists were considered: a cyclist riding individually, a medium-sized group formed by four cyclists, and a large-sized group formed by ten cyclists. For medium and large groups, the effect of the two-abreast or in-line group configuration was also analysed.

Data collection was conducted using instrumented bicycles and a group of sport cyclists who rode along five rural road segments. The road segments have different geometric and traffic characteristics; however, this study is focused on the analysis of how factors related to road geometric characteristics influence the overtaking manoeuvre to cyclists. To achieve that, multiple regression models for the three different sizes of cyclist groups were developed. The geometric characteristics of the road considered as factors in the models were the road centre-line type, the horizontal alignment, and the lane and shoulder widths. The invasion of the opposing lane during the overtaking manoeuvre was also considered as a factor in the models, due its relationship with the road cross section. The effect of in-line or two-abreast configuration was also examined for groups of cyclists.

The results of this study allow for the identification of differences in the overtaking manoeuvre to groups of cyclists regarding their size and configuration. Regarding lateral clearance, when overtaking one cyclist riding individually, the lane and shoulder widths and the invasion of the opposing lane had a significant effect, such that an increase in lane and shoulder widths increased the lateral clearance. This lateral clearance was higher when the opposing lane was invaded. Similar results were obtained for the medium-sized group formed by four cyclists. For the large group of cyclists (10 cyclists) the factors with a significant effect on the lateral clearance were the road centre line and the invasion of the opposing lane, generating higher clearances of overtaking manoeuvres performed in a dashed line (an increment of up to 0.11 m) and with invasion of the opposing lane. Regarding overtaking vehicle speed, when one cyclist rode individually the speed was higher for the wider lane and shoulders and for overtaking manoeuvres performed in a dashed road centre line. For a medium-sized group of cyclists, the overtaking vehicle speed increased in the same conditions, but in this case the invasion of the opposing lane also had a significant effect, increasing the speed when the opposing lane was invaded. When a larger group of cyclists was overtaken, only the road centre line had a significant effect on the overtaking vehicle speed, increasing the speed when the overtaking was performed in a dashed line. The in-line or two-abreast configuration in which the group of cyclists ride has a significant effect on both lateral clearance and speed of the overtaking vehicle. This significant effect was found for medium and large groups of cyclists, such that when cyclists rode in-line, higher lateral clearances and higher overtaking vehicle speeds were obtained, increasing for the large group up to 0.17 m and 5.83 km/h when cyclists rode in-line.

This study aims to fit explanatory models considering only geometric road variables, so as to reflect the magnitude and sign of the influence of each geometric variable on lateral clearance and overtaking vehicle speed. With simple models, such as multiple linear regression models, the designer can obtain an overview of the importance of these variables in the future road design (or improvement of existing roads). The results show that considering road geometric factors only, a higher proportion of the variability of overtaking manoeuvres to cyclists is explained when they ride individually, whereas when they ride in larger groups, geometric factors explain a smaller part of the variability. This indicates that for larger groups of cyclists, other factors have an important role in this phenomenon.

The results of this study are limited to level two-lane rural roads with the observed geometric and traffic characteristics. Is evident that more factors influence the overtaking manoeuvre to cyclists on two-lane rural roads. Interactions between drivers and cyclists are a complex phenomenon, involving many factors, such as infrastructure, traffic, and human factors. In this study, however, only factors related to road geometry have been considered, analysing their influence on speed and lateral clearance during the overtaking manoeuvre. By including only geometric variables, the model fits, especially for large groups of cyclists, were not high. Therefore, further research is necessary to incorporate the other variables related to the phenomenon (traffic factor and human factor) in the models, in order to offer a more exhaustive view of the phenomenon. However, the models obtained in this study provide insight into which factors have a significant effect on the lateral clearance and speed of overtaking vehicles as a function of the size of the group of cyclists and the sign of this effect. Additionally, a novelty of this study is that it considers cyclists riding in groups, as this is a reality on Spanish rural two-lane roads. These results can be transformed into recommendations and proposals to improve the safety of all users of two-lane rural roads. To generalise the results to other countries, naturalistic data should be taken from interactions between drivers and cyclists and checked as to whether the behaviour of both is similar.

As further research, it is proposed to fit a more complete model using a Bayesian approach in order to obtain the distributions of the parameters with a strong effect on the lateral clearance and the overtaking vehicle speed. Furthermore, the effect of road geometry can be related to the risk perception of cyclists in overtaking manoeuvres to increase the safety perception of these road users.

## Figures and Tables

**Figure 1 ijerph-19-09302-f001:**
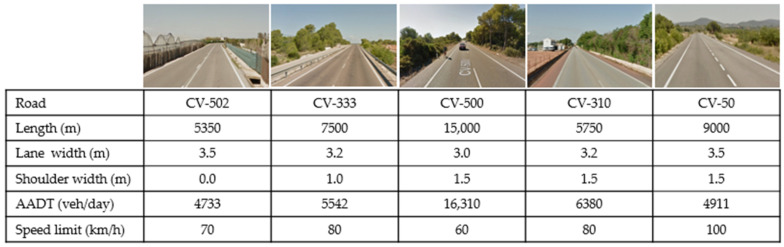
Characteristics of the five two-lane rural road segments where the study was developed.

**Figure 2 ijerph-19-09302-f002:**
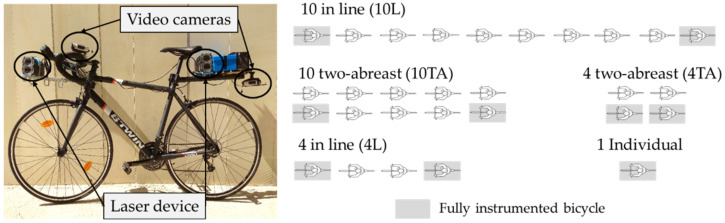
Instrumented bicycle and cyclists’ group configurations.

**Figure 3 ijerph-19-09302-f003:**
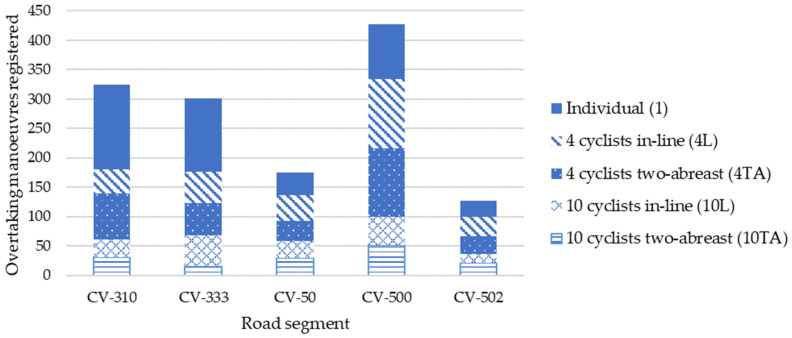
Overtaking manoeuvres registered per road segment and cyclist group configuration.

**Figure 4 ijerph-19-09302-f004:**
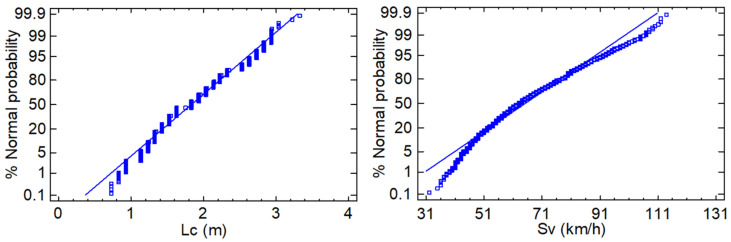
Normal probability plot for observed lateral clearance and overtaking vehicle speed.

**Figure 5 ijerph-19-09302-f005:**
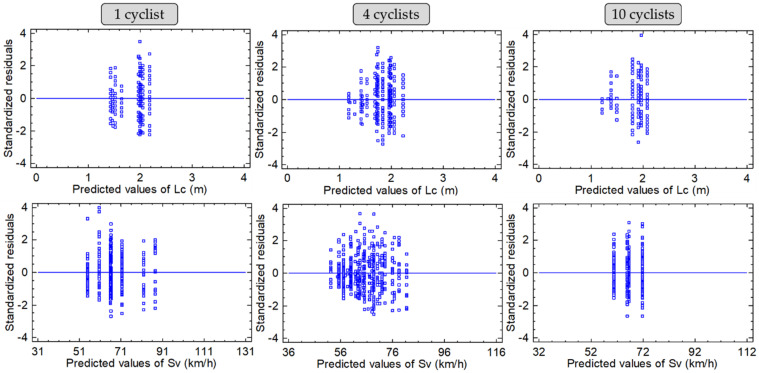
Standardized residual plots for lateral clearance (**upper**) and overtaking speed (**bottom**).

**Table 1 ijerph-19-09302-t001:** Variables related to road geometry and configuration of group of cyclists.

Variable	Code	Variable Type	Values
Lane width	Lw	quantitative discrete	3, 3.2 and 3.5
Shoulder width	Sw	quantitative discrete	0, 1 and 1.5
Lane + Shoulder width	LSw	quantitative discrete	3.5, 4.2, 4.5, 4.7 and 5
Centre-line	CL	qualitative	Solid line (SL), Dashed line (DL)
Horizontal alignment	HA	qualitative	Tangent (T), Right curve (RCur), Left curve (LCur)
Speed limit	SL	quantitative discrete	60, 70, 80 and 100 km/h
Opposing lane invasion	Inv	qualitative	Invasion (Inv), No invasion (NInv)
Configuration	Conf	qualitative	In-line (IL), Two-abreast (TA)

**Table 2 ijerph-19-09302-t002:** Lateral clearance and overtaking vehicle speed results from field data.

		1 Cyclist					4 Cyclists					10 Cyclists				
			Lateral Clearance (m)		Overtaking Vehicle Speed (km/h)			Lateral Clearance (m)		Overtaking Vehicle Speed (km/h)			Lateral Clearance (m)		Overtaking Vehicle Speed (km/h)	
Variables	Values	*N*	Mean	SD	Mean	SD	*N*	Mean	SD	Mean	SD	*N*	Mean	SD	Mean	SD
Lane width	3	*94*	1.63	0.40	52.71	8.38	*235*	1.70	0.44	57.15	8.94	*99*	1.77	0.50	62.79	8.11
	3.2	*269*	1.87	0.46	68.30	13.68	*227*	1.84	0.46	66.76	15.80	*129*	1.89	0.50	67.62	15.73
	3.5	*66*	2.00	0.50	75.58	18.39	*142*	1.93	0.44	71.51	15.93	*94*	1.94	0.49	67.30	14.31
Shoulder width	0	*28*	1.89	0.45	66.14	15.33	*63*	1.80	0.40	66.10	14.62	*36*	1.90	0.46	63.39	11.38
	1	*125*	1.95	0.50	66.09	16.28	*108*	1.93	0.47	65.08	15.57	*68*	1.96	0.47	70.85	16.12
	1.5	*276*	1.78	0.45	65.95	15.20	*433*	1.78	0.46	63.61	14.59	*218*	1.83	0.52	64.98	12.64
Lane + Shoulder width	3.5	*28*	1.89	0.45	66.14	15.33	*63*	1.80	0.40	66.10	14.62	*36*	1.90	0.46	63.39	11.38
	4.2	*125*	1.95	0.50	66.09	16.28	*108*	1.93	0.47	65.08	15.57	*68*	1.96	0.47	70.85	16.12
	4.5	*94*	1.63	0.40	52.71	8.38	*235*	1.70	0.44	57.15	8.94	*99*	1.77	0.50	62.79	8.11
	4.7	*144*	1.81	0.42	70.22	10.63	*119*	1.76	0.44	68.28	15.92	*61*	1.81	0.53	64.02	14.58
	5	*38*	2.07	0.53	82.53	17.50	*79*	2.04	0.44	75.82	15.69	*58*	1.96	0.52	69.72	15.45
Speed limit	60	*94*	1.63	0.40	52.71	8.38	*235*	1.70	0.44	57.15	8.94	*99*	1.77	0.50	62.79	8.11
	70	*28*	1.89	0.45	66.14	15.33	*63*	1.80	0.40	66.10	14.62	*36*	1.90	0.46	63.39	11.38
	80	*269*	1.87	0.46	68.30	13.68	*227*	1.84	0.46	66.76	15.80	*129*	1.89	0.50	67.62	15.73
	100	*38*	2.07	0.53	82.53	17.50	*79*	2.04	0.44	75.82	15.69	*58*	1.96	0.52	69.72	15.45
Centre line	Solid	*222*	1.78	0.44	61.87	15.05	*370*	1.76	0.45	60.54	12.49	*187*	1.79	0.48	63.60	11.52
	Dashed	*207*	1.90	0.49	70.43	14.75	*234*	1.88	0.46	69.82	16.28	*135*	1.97	0.51	69.42	15.29
Horizontal alignment	Tangent	*374*	1.82	0.45	65.70	15.38	*512*	1.79	0.45	63.58	14.44	*282*	1.87	0.49	66.51	13.55
	Right curve	*32*	1.87	0.65	70.50	16.32	*41*	1.83	0.41	66.37	17.26	*16*	1.96	0.54	67.75	16.20
	Left curve	*23*	2.11	0.43	64.65	15.93	*51*	1.97	0.50	67.92	15.59	*24*	1.79	0.57	59.33	9.20
Opposing lane invasion	No invasion	*147*	1.48	0.29	63.65	14.61	*95*	1.45	0.35	59.93	12.43	*39*	1.37	0.34	67.51	14.86
	Invasion	*282*	2.03	0.44	67.23	15.82	*509*	1.88	0.44	64.92	15.06	*283*	1.93	0.48	65.84	13.34
Configuration	In-line						*288*	1.87	0.45	64.97	13.43	*172*	1.92	0.48	69.02	13.65
	Two-abreast						*316*	1.76	0.46	63.38	15.89	*150*	1.81	0.52	62.62	12.56
Total		*429*	1.84	0.47	66.00	15.49	*604*	1.81	0.46	64.14	14.78	*322*	1.87	0.50	66.04	13.52

**Table 3 ijerph-19-09302-t003:** General regression models for lateral clearance and overtaking vehicle speed considering cyclists riding individually or in groups.

GM for Lc	Lc = −0.2372 + 0.1086 × Individual + 0.4670 × Lane width + 0.1127 × Shoulder width + 0.5025 × Invasion						
Parameters	Coeficient	Error	*t* stat	*p*-value	Upper 95%	Lower 95%	Model statistics
Intercept	−0.2372	0.2572	−0.9220	0.3565	−0.7413	0.2670	R^2^ = 0.2045
Individual	0.1086	0.0253	4.2895	<0.0001	0.0590	0.1583	Adjusted R^2^ = 0.2022
Lane width	0.4670	0.0736	6.3413	<0.0001	0.3227	0.6114	Observations No. = 1355
Shoulder width	0.1127	0.0299	3.7741	<0.0001	0.0542	0.1712	DW = 1.7196 (*p* < 0.0001)
Invasion	0.5025	0.0297	16.8963	0.0002	0.4442	0.5607	F = 86.78 (*p* < 0.0001)
**GM for Sv**	**Sv = −35.0928 + 29.7949 × Lane width + 6.6754 × Shoulder width − 6.1133 × Solid line**						
Parameters	Coeficient	Error	*t* stat	*p*-value	Upper 95%	Lower 95%	Model statistics
Intercept	−35.0928	8.4854	−4.1357	<0.0001	−51.7239	−18.4617	R^2^ = 0.1717
Lane width	29.7949	2.4200	12.3121	<0.0002	25.0518	34.5379	Adjusted R^2^ = 0.1699
Shoulder width	6.6754	0.9426	7.0816	<0.0003	4.8279	8.5230	Observations No. = 1355
Solid line	−6.1133	0.7865	−7.7729	<0.0004	−7.6548	−4.5718	DW = 1.3966 (*p* < 0.0001)
							F = 93.38 (*p* < 0.0001)

**Table 4 ijerph-19-09302-t004:** Regression models for lateral clearance considering the different cyclist groups.

M1: 1 Individual	Lc = −0.0852 + 0.4454 × Lane width + 0.1144 × Shoulder width + 0.5383 × Invasion						
Parameters	Coeficient	Error	*t* stat	*p*-value	Upper 95%	Lower 95%	Model statistics
Intercept	−0.0855	0.4918	−0.1731	0.8626	−1.0519	0.8816	R^2^ = 0.3233
Lane width	0.4454	0.1437	3.0995	0.0021	0.1630	0.7279	Adjusted R^2^ = 0.3185
Shoulder width	0.1144	0.05413	2.1142	0.0351	0.0080	0.2208	Observations No. = 429
Invasion	0.5383	0.0415	12.9724	<0.0001	0.4567	0.6198	DW = 1.6434 (*p* = 0.0001)
							F = 67.67 (*p* < 0.0001)
**M2: 4 Cyclists**	**Lc = −0.7048 + 0.2348 × In-line + 0.5454 × Lane width + 0.1654 × Shoulder width + 0.5387 × Invasion**						
Parameters	Coeficient	Error	*t* stat	*p*-value	Upper 95%	Lower 95%	Model statistics
Intercept	−0.7048	0.3629	−1.9422	0.0526	−1.4176	0.0079	R^2^ = 0.2164
In-line	0.2348	0.0350	6.7043	<0.0001	0.1660	0.3035	Adjusted R^2^ = 0.2116
Lane width	0.5454	0.1027	5.3082	<0.0001	0.3436	0.7471	Observations No. = 604
Shoulder width	0.1654	0.0426	3.8802	0.0001	0.0817	0.2491	DW = 1.7960 (*p* = 0.0061)
Invasion	0.5387	0.0489	11.0174	<0.0001	0.4426	0.6347	F = 41.35 (*p* < 0.0001)
**M3: 10 Cyclists**	**Lc = 1.320 + 0.1695 × In-line − 0.1125 × Solid-line + 0.5930 × Invasion**						
Parameters	Coeficient	Error	*t* stat	*p*-value	Upper 95%	Lower 95%	Model statistics
Intercept	1.320	0.0954	13.8432	<0.0001	1.1324	1.5076	R^2^ = 0.1789
In-line	0.1695	0.0526	3.2227	0.0014	0.0660	0.2729	Adjusted R^2^ = 0.1712
Solid-line	−0.1125	0.0526	−2.1413	0.0330	−0.2159	−0.0091	Observations No. = 322
Invasion	0.5930	0.0805	7.3659	<0.0001	0.4346	0.7514	DW = 1.8404 (*p* = 0.0762)
							F = 23.10 (*p* < 0.0001)

**Table 5 ijerph-19-09302-t005:** Regression models for overtaking vehicle speed considering the different cyclist groups.

M4: 1 Individual	Sv = −117.206 + 53.936 × Lane width + 10.687 × Shoulder width − 5.681 × Solid-line						
Parameters	Coefficient	Error	*t* stat	*p*-value	Upper 95%	Lower 95%	Model statistics
Intercept	−117.206	17.1352	−6.8401	<0.0001	−150.886	−83.526	R^2^ = 0.2802
Lane width	53.936	4.9385	10.9216	<0.0001	44.229	63.643	Adjusted R^2^ = 0.2752
Shoulder width	10.687	1.8128	5.8950	<0.0001	7.123	14.250	Observations No. = 429
Solid-line	−5.681	1.3228	−4.2949	<0.0001	−8.281	−3.081	DW = 1.4353 (*p* < 0.0001)
							F = 55.16 (*p* < 0.0001)
**M5: 4 Cyclists**	**Sv = −49.413 + 3.008 × In-line + 31.925 × Lane width + 7.501 × Shoulder width − 5.425 × Solid-line + 4.869 × Invasion**						
Parameters	Coefficient	Error	*t* stat	*p*-value	Upper 95%	Lower 95%	Model statistics
Intercept	−49.413	12.6885	−3.8943	0.0001	−74.333	−24.494	R^2^ = 0.2173
In-line	3.008	1.1350	2.6499	0.0083	0.779	5.237	Adjusted R^2^ = 0.2108
Lane width	31.925	3.5483	8.9974	<0.0001	24.956	38.894	Observations No. = 604
Shoulder width	7.501	1.3793	5.4384	<0.0001	4.793	10.210	DW = 1.4203(*p* < 0.0001)
Solid-line	−5.425	1.2161	−4.4613	<0.0001	−7.813	−3.037	F = 33.21 (*p* < 0.0001)
Invasion	4.869	1.5908	3.0607	0.0023	1.745	7.993	
**M6: 10 Cyclists**	**Sv = 65.924 + 5.831 × In-line − 5.162 × Solid-line**						
Parameters	Coefficient	Error	*t* stat	*p*-value	Upper 95%	Lower 95%	Model statistics
Intercept	65.924	1.4138	46.6304	<0.0001	63.142	68.705	R^2^ = 0.0912, Adj. R^2^ = 0.0855
In-line	5.831	1.4535	4.0114	0.0001	2.971	8.690	Observations No. = 322
Solid-line	−5.162	1.4694	−3.5132	0.0005	−8.053	−2.271	DW = 1.4543 (*p* < 0.0001)
							F = 16.00 (*p* < 0.0001)

## Data Availability

Data and models that support the findings of this study are available from the corresponding author upon reasonable request.

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
