# Peer review of "Evaluation of the Influence of Road Geometry on Overtaking Cyclists on Two-Lane Rural Roads"

_ijerph, 2022, doi:10.3390/ijerph19159302_

Round 1

Reviewer 1 Report

Evaluation of the influence of road geometry on overtaking cyclists on two-lane rural roads

A brief summary

The paper analyses how road geometry influences overtaking manoeuvre performance. Field data of overtaking manoeuvres were collected using instrumented bicycles, riding along different rural road segments, and considering individual, medium and large groups of cyclists. The safety variables that are considered characterize overtaking manoeuvre are overtaking vehicle speed and lateral clearance.

Comments

1.     The idea of the research is interesting and presents enough novelty.

2.     The paper should attract an audience in the field of road safety.

3.     The paper fits the topics of the journal.

4.     The proposed method seems to be innovative and contains well-known hints of originality.

Weakness of the paper:

1)    Authors should anticipate some results in the abstract to capture the readers’ curiosity.

2)    The results of the paper should be better highlighted.

3)    The most significant numerical results should also be reported in the conclusions.

4)    It would be useful to add in figure 1, some data on the geometric characteristics of the route of the roads examined (length, minimum radius, maximum slope, maximum absolute variation in elevation,% heavy traffic, etc.).

5)    I propose to perform preliminarily a Shapiro-Wilk test of normality on the data.

6)    Lines 254-260: "the model developed for individual cyclists presents the highest value (adjusted R2 = 0.3185);"

7)    The results are presented by the authors in a promising way and a reader unfamiliar with the statistics might be captivated by these very interesting conclusions. I regret to point out to the authors that the values of R2 obtained are much less than 1 and, in my opinion, leave no possibility that the proposed models can be used to estimate unknown greatness. Therefore, I invite the authors to examine in depth the statistical models proposed tath they presented in this manuscript, considering that they are not applicable having such a low value of R2.

8)    … no more weaknesses!

The overall merit of presented research works and findings can be taken into consideration for publishing after incorporating the above suggestions.

Author Response

Attached is the response to the review.

Reviewer 2 Report

Abstract - The first sentence is redundant.

The abstract should include some quantitative findings (e.g., p-values, effect size, coefficients, etc.). Also include the sample size and in the end, include a sentence on applications of the study.

Literature review - A brief table providing the key studies on overtaking behavior of cyclists will benefit the readers and will help in bringing out clarity about the research motivation. At present, the motivation of the study appears weak.

Table 1 - Write the full form of quant. for better understanding.

More details on data collection is required - such as time of data collection, surrounding traffic density, road features, etc.

Proper justification for selecting regression based modelling is required.

As this study was conducted in Spain, how can the findings be generalized to other countries?

At the end of discussion, the authors should provide a subsection on "Research contributions" where they can highlight the contributions of the study in bullets for easy understanding of the readers.

Author Response

Attached is the response to the review.

Round 2

Reviewer 1 Report

The authors addressed the reviewers' comments and the paper is improved. I propose to publish the paper.

Reviewer 2 Report

The authors have significantly improved the quality of their manuscript. I do not have any further suggestions. Best wishes to the authors!